# Effect of Individualized Coaching at Home on Quality of Life in Subacute Stroke Patients

**DOI:** 10.3390/ijerph20105908

**Published:** 2023-05-22

**Authors:** Rodeline Telfils, Axelle Gelineau, Jean-Christophe Daviet, Justine Lacroix, Benoit Borel, Emna Toulgui, Maxence Compagnat, Stéphane Mandigout

**Affiliations:** 1HAVAE UR20217, University of Limoges, F-87000 Limoges, France; 2Department PMR, CHU Limoges, F-87000 Limoges, France; 3Department of Physical Medicine and Rehabilitation, Sahloul University Hospital, Sousse 4054, Tunisia

**Keywords:** stroke, physical activity, quality of life, home

## Abstract

**Background**: Stroke causes psychological disorders and cognitive impairments that affect activities of daily living and quality of life. Physical activity (PA) in stroke recovery is beneficial. The benefits of PA on quality of life after stroke are less documented. The aim of the study was to evaluate the effect of a home-based PA incentive program at home in post-stroke patients in the subacute phase on quality of life. **Methods**: This is a prospective, randomized, single-blind, and monocentric clinical trial. Eighty-three patients were randomly assigned to either an experimental group (EG; *n* = 42) or to a control group (CG; *n* = 41). The experimental group followed a home-based PA incentive program for 6 months. Three incentive methods were used: daily monitoring with an accelerometer, weekly telephone calls, and home visits every three weeks. Patients were evaluated before intervention (T0) and after intervention (T1) at 6 months. The control group was a non-intervention group receiving usual care. The outcome was the quality of life with the EuroQol EQ-5D-5L evaluated at baseline and 6 months post-intervention. **Results**: The mean age was 62.2 years ± 13.6 with a post-stroke time of 77.9 ± 45.1 days. The mean values of the utility index (EQ-5D-5L) in the control group and experimental group at T1 were 0.721 ± 0.207 and 0.808 ± 0.193, respectively (*p* = 0.02). **Discussion**: Our study shows a significant difference in the Global QOL index (EQ-5D-5L) between the two groups of subacute stroke patients after 6 months of the individualized coaching program, which combined home visits and weekly telephone calls.

## 1. Introduction

Stroke is a major public health problem because of its frequency and the resulting physical and cognitive disabilities [1]. More than 150,000 new cases are reported each year in France, including 40,000 deaths and 30,000 cases of severe disability [2,3]. Overall, the consequences of stroke lead to a decrease in the patient’s physical capacities. Fatigue and Quality of Life (QOL) are negatively impacted [4,5].

Physical activity (PA) appears to have a favorable effect on QOL in the general population [6]. The benefits of PA after stroke are numerous. Physical activity improves cognitive function and walking capacity [7]. The benefits of PA on QOL after stroke are less documented, and studies have mainly been conducted in chronic patients. In subacute patients, there are relatively few results. To the best of our knowledge, only four studies [8,9,10,11] have evaluated the effect of a home PA program in the subacute phase on QOL, and only two of them used the EQ-5D-5L questionnaire [9,10]. They all seem to show benefits for the post-stroke patient.

Unfortunately, the authors who studied the durability of these benefits showed that they were not maintained over time after the PA program was stopped and that most PA programs were carried out in medicalized structures with a standardized PA program [8,12]. In this context, it is important to find strategies to maintain the benefits of PA. The literature shows that the establishment of support teams, which intervene at home after hospital discharge, is a beneficial strategy for dealing with these types of difficulties [13].

As far as we know, up until now, no study has evaluated the impact of a PA incentive program based on the measurement of spontaneous PA using three incentive strategies: accelerometer, telephone calls, and home visits. In this context, the aim of this study was to evaluate the effect of a 6-month home-based PA incentive program on the QOL of patients in the subacute phase of stroke recovery.

## 2. Materials and Methods

### 2.1. Study Design, Setting, and Participants

This was a monocentric study that employed a prospective randomized controlled clinical trial design with two groups—an experimental group (EG) and a control group (CG). The observer was blinded. The study protocol, including details of randomization and intervention, as well as the statistical analysis plan, was registered at http://ClinicalTrials.gov (accessed on 4 April 2013) (NCT01822938) and published previously [14]. This study was conducted in Limoges, France, with participants recruited from the physical medicine and rehabilitation services at the hospital center. The study adhered to the Good Clinical Practice Guidelines of the Declaration of Helsinki and followed the CONSORT 2010 statements for randomized trials of non-pharmacological treatments. Ethical approval was obtained from institutional committees (RCB: 2012-A01456-37) and potential participants with a first stroke who met the inclusion criteria (age ≥ 18 years; first ischemic or hemorrhagic stroke within <6 months; Modified Functional Ambulatory Categories (FAC) ≥ 2; registered with the French social security system, patient monitoring by the post-stroke interventional mobile team (HEMIPASS)) were screened. Exclusion criteria included pre- or post-stroke physical disorders limiting gait skills, cognitive disorders impeding comprehension of PA education—i.e., aphasia, as indicated by a Boston Diagnostic Aphasia Examination score < 2, uncontrolled hypertension, inability to complete questionnaires, cardiopulmonary pathology preventing effort completion, involvement in other research, legal guardianship, and pregnancy.

Assessments were performed at baseline (T0), 6 months (T1), and 12 months (T2), the first patient was enrolled on 29 April 2013, and the last end-of-trial consultation was conducted on 17 January 2018. Each patient participated in the study for 12 months, and the total duration of the research was 4 years, 8 months, and 18 days.

All subjects who met the eligibility criteria provided written informed consent in accordance with the study protocol. Using a secure internet connection, participants were randomly allocated (in blocks of variable size) to either an EG that received regular personalized coaching on PA or a CG that received standard care. The randomization process was guaranteed through an individual user ID and password, as well as encrypted data transmission to the randomization platform of the Clinical Research Unit of the Limoges University Hospital.

### 2.2. Interventions

Following inclusion, a PA advice program was designed and implemented by a PA therapist for all eligible participants according to the recommendations of the French Society of Physical Medicine and Rehabilitation and the French Neurovascular. The presentation included an educational diagnosis (benefits of an active lifestyle in preventing stroke recurrence), selection of PA, and definition of goals [15]. Usual care was provided for all eligible participants for 12 months after hospital discharge, including outpatient therapy and medical appointments at 1, 6, and 12 months. The EG underwent an incentive coaching program, which involved monitoring, weekly phone calls, and a home visit every 3 weeks for 6 months. An activity tracker (AT) (SenseWear Armband, BodyMedia, Pittsburgh, PA, USA) was used to monitor PA at home. The participants were asked to wear the AT when they woke up and to remove it before going to bed. This sensor was selected due to its frequent utilization in activity analysis and its unique feature of activation upon contact with the wearer’s skin. The data obtained correspond to AT wearing time. The EG participants were routinely monitored through the completion of a daily chart to measure the subjective perception of PA, weekly phone calls to encourage regular PA and inquire about the PA measuring device, and home visits every three weeks to receive participant feedback on PA level (step counts [SC]) and set objectives for the next visit. The aim of this program was to encourage participants to meet the PA recommendations without offering them specific sessions but rather encouraging them to maintain or have an active lifestyle.

The AT recorded and analyzed physiological parameters using algorithms to calculate total and active Energy Expenditure (EE), number of steps, duration (expressed in minutes), and intensity of PA (expressed in metabolic equivalent of task MET). The configuration of the sensor in relation to PA intensity was defined as follows: low (<3 METs), moderate (≥3 to 5.9 METs), vigorous (≥6 to 8.9 METs), and very vigorous (≥9 METs). In contrast, the CG corresponds to a non-intervention group, receiving only usual care without PA monitoring, home visits, or telephone calls. Additionally, the CG did not receive further coaching on PA during the 6-month follow-up (T1).

### 2.3. Measurements

For outcome measurements, all subjects were evaluated first after hospital discharge (T0) and then immediately after the 6-month intervention (T1). All assessments were conducted by the same medical doctor and professional PA therapist in the hospital. The main outcome was published in a previous study (Papier TICADOM). In this paper, we appraised the effect of the program on QOL as evaluated with EQ-5D-5L at T1. EQ-5D-5L is a standardized questionnaire that measures health status through five dimensions: mobility, self-care, usual activities, pain and discomfort, and anxiety/depression. For each of the dimensions, there are five levels coded from 1 to 5; by combining the 5 dimensions, a 5-digit number is obtained, ranging from 11,111 (no problem) to 55,555 (extreme problem). The number corresponding to health status was then converted into an index value, and delta EQ-5D-5L between T0 and T1 was calculated [16]. Other outcomes were framed in terms of the Barthel Index (BI), Motor Demeurisse Index (MDI), Functional Ambulation Categories (mFAC), and 6-Minute Walk Test (6MWT). BI is a scale evaluating the dependence and independence of patients in which a score of 0 represents total dependence, and 100 represents complete independence [17]. MDI is a validated stroke scale [18]. It scores, in the sitting position, the anterior elevation of the upper limb, elbow flexion, thumb-index terminal grip, hip flexion, knee extension, and foot dorsiflexion. The assessment yields a motor score out of 100 for the upper limb and out of 100 for the lower limb; each of these scores is divided by two to obtain an overall score out of 100. “100” indicates good motor status, and 0 indicates poor motor status. mFAC allows functional assessment of walking [19]. It classifies ambulation from 0 (nonfunctional ambulation) to 8 (independent ambulation); 6MWT is an objective measurement of exercise capacity in people with significant disabilities (American Thoracic Society ATS) [20]. The 6-min walk distance is expressed in meters.

### 2.4. Statistical Analysis

Data were analyzed with R version 4.1.3 (Vienna, Austria). All analyses were performed as per protocol. The normality of the variables analyzed was tested using the Shapiro–Wilk test. Descriptive statistics were used to characterize demographics, performance, and clinical characteristics for each group. Continuous data were tested using Student *t*-tests or Wilcoxon tests. The continuous outcomes of 6MWT, FAC, Barthel index, and Motricity index were analyzed by Repeated measures analysis of variance (ANOVA). For the EG, correlations between PA level (Average daily PA, expressed in number of steps, Total Energy Expenditure (TEE) (Kcal), and duration of moderate PA in minutes) and EQ-5D-5L were tested at T1 using Spearman’s test or linear correlation coefficient plus *t*-test. The level of significance was set at 0.05.

## 3. Results

The population consisted of 84 post-stroke patients. One patient withdrew his consent and was excluded from the analyses. The number of subjects participating in the study was 83 (EG, *n* = 42 and CG, *n* = 41). At baseline, there were no significant differences in group variables or outcome measures. The Shapiro–Wilk test showed that all variables, except for the 6MWT at T0, did not follow a normal distribution (*p*-value < 0.05. Thus, non-parametric tests were used for the subsequent analyses.

### 3.1. Descriptive Analysis of the Sample

The characteristics of our sample population are detailed in Table 1. The mean age was 62.2 ± 13.6 years, with a post-stroke time lapse of 77.9 ± 45.1 days (Table 1).

At T0, patients had a high autonomy level with a mean BI score of 95.1 ± 8.9 (out of 100), an MDI score of 87.5 ± 15.4 (out of 100), and a modified FAC score of 6.5 ± 1.4 (out of 8). The mean EQ-5D5L index was 0.723 ± 0.200 (Table 2).

The mean distance covered during the 6MWT was 364.5 ± 151.3 m at T0 (Table 2).

### 3.2. Effect of the Program on the QOL

The EQ-5D-5L index at T1 was 0.721 ± 0.207 for CG and 0.808 ± 0.193 for EG. EG increased the EQ-5D-5L index by 0.062 after the program, and CG increased by 0.014.

The Wilcoxon test reported a significant difference in the EQ-5D5-L index at 6 months (T1) between CG and EG (*p* = 0.02). (Figure 1).

Table 3 shows significant differences in the distribution of responses in the mobility for the total group (*n* = 83), usual activities, and pain/discomfort dimensions of the EQ-5D-5L (*p* = 0.01; 0.03; 0.01, respectively).

### 3.3. Effects on Other Outcomes

The repeated measures analysis of variance reports an interaction of the Group and Time (*p* = 0.0011) factors for the criterion of walking distance at the 6MWT. A significant increase in 6MWT was achieved for the EG between T0 and T1 (18%, *p* < 0.001) (Table 4).

The evolution of the mFAC score between T0 and T1, evaluated from the analysis of variance, differed between our two groups (*p* = 0.036). The post-hoc test confirmed the significant effect of the incentive program on ambulation ability between T0 and T1 (*p* = 0.01), but only for the EG (Table 4).

Delta EQ-5D-5L between T0 and T1 was significantly and negatively correlated with TEE (r = −0.42; *p* = 0.009) and moderate intensity PA per day (r = −0.37; *p* = 0.02). Correlation analysis showed that there was no significant relationship between the number of steps per day and EQ-5D-5L (r = −0.3; *p* = 0.05).

## 4. Discussion

### 4.1. Effect of PA Incentive Program on QOL

The objective of the study was to evaluate the impact of a home-based PA incentive program on QOL with the EQ-5D-5L in the subacute post-stroke patient. The major result of our study shows a significant difference between the EG and the CG (*p* = 0.02) in the EQ-5D-5L index at the end of the intervention period (T1).

The findings of our study are consistent with those of Chaiyawat et al., prospective RCT involving 60 patients with recent ischemic stroke [10]. The study suggested a 6-month individualized home exercise program administered by a physical therapist once a month, with sessions of approximately 1 h, for 6 months. The PA program included various rehabilitation methods such as passive exercise, active exercise, resistance exercise, and activities of daily living. At the end of their intervention, the mean (SD) utility index (EQ-5D-5L) in the intervention and control groups was 0.9 ± 0.02 and 0.7 ± 0.04, *p* = 0.03, respectively. The percentage increase in the EQ-5D-5L index of our patients was less than in the study by Chaiyawat et al. [10], in which their PA protocol resulted in an 84% increase in EQ-5D-5L index in the EG and CG. The level of autonomy may explain the difference in percentages of EQ-5D-5L improvement between the two studies. Indeed, the BI score of our patients at T0 (EG = 94.8 ± 9.6 and CG = 94.7 ± 8.5) was 63.1 and 64.2 points higher compared with the patients in the Chaiyawat et al. study (EG = 31.7 and CG = 33.2) [10].

Another factor that could have confounded the results was the improved physical level of the participants. In our main publication [21], we showed that over 6 months of the program, essentially based on an incentive to increase their level of daily activity, the number of steps had significantly increased. However, TEE and TAEE remained relatively stable [21].

The PA program increased the walking distance covered at the 6MWT by 11.7 m in the CG and by 68. 9m in the EG. The evolution of the performance at the 6MWT between T0 and T1, evaluated from an analysis of variance, differed between our two groups. Post-hoc tests showed a significant 6MWT increase for the EG (18%, *p* < 0.001) after the program. These results should be taken with great caution. Indeed, in our main study [21], we did not show a difference at T1 between the two groups when the statistical analysis was performed on intention-to-treat. However, the primary endpoint was calculated based on a 30% increase in 6MWT. This result could not be achieved due to the low level of impairment of our program participants (Barthel index > 94). Our results are in agreement with the study by Duncan et al. [22]. This study proposed a home-based PA program, three times a week, with sessions of approximately 1.5 h, for 12 weeks. At the end of the intervention, the PA program increased the walking distance at the 6MWT by 59 m for the patients who followed the home program and by 35 m for the control patients. We show in this secondary analysis that, despite a high level of heterogeneity and a low level of impairment, our incentive program, consisting of individualized follow-up, significantly improved the quality of life and walking capacity of our participants.

### 4.2. Effect of PA Incentive Program on QOL Domains

Generally, QOL questionnaires propose not only a global index but also an index concerning several dimensions. The EQ5D-5L evaluates mobility, self-care, usual activity, pain and discomfort, and anxiety/depression. Despite the improvement of the global index of QOL in the EG only, the chi-square analysis of the frequency distribution in each of the five dimensions over the five levels shows a significant difference in mobility, usual activity, pain and discomfort, and anxiety/depression for the entire sample, whereas no difference was found when the two groups were separated. A trend was observed in the EG with usual activity (*p* = 0.10). These results are rarely found in the literature on this population.

Similarly, PA programs have been reported to improve patients’ QOL, as shown in the study by Mead et al. [23]. The authors used the MOS SF-36 questionnaire to assess QOL and, at the end of the PA program, reported a significantly increased the score in 5 domains (limitations due to physical condition, psychological health, limitations due to psychological condition, perceived health, and vitality) out of the 8 MOS SF-36 domains.

It is, therefore, difficult to determine the elements that can explain our results. We suppose that the explanation could come from the size of our sample, which was not adapted to this criterion.

### 4.3. Relation between QOL Assessment and Level of Physical Activity during Daily Living

Furthermore, our study shows that the delta of EQ-5D-5L between T0 and T1 was significantly negatively correlated with TEE (r = −0.42; *p* = 0.009) and TAEE per day (r = −0.37; *p* = 0.02). This indicates that when TEE and TAEE increase, EQ-5D-5L decreases. Specifically, the participants with the highest PA practice presented the most pronounced improvement in QOL.

### 4.4. Limits of the Study

A recruitment bias was detected during inclusion. The participants were mainly patients with a high level of autonomy (BI of >90 points), and motricity (mFAC > 6), and patients with very severe stroke do not wish to be included in this study. It would be interesting to evaluate whether this type of program could be feasible and provide benefits for far less autonomous patients.

In our study, it is difficult to estimate the contribution of physical activity in improving the quality of life of participants. Other factors may confound the results, such as the regular presence of health professionals from the HEMIPASS team in the homes of EG patients (every 3 weeks). This could easily help to improve patients’ quality of life. On this subject, it is important to recall that all patients included in the study benefited from the monitoring of the HEMIPASS mobile team. EG patients had more regular monitoring. We assume that the improvement in quality of life in our sample can be explained by the regular and individualized monitoring of the incentive program.

## 5. Conclusions

This study investigated the impact of a home-based PA incentive program in subacute post-stroke patients on QOL using the EQ-5D-5L questionnaire. The PA incentive program improved QOL in subacute post-stroke patients. The increase was significantly different (*p* = 0.02) between the two groups after the intervention. In addition, the PA incentive program improved functional capacity and walking distance in the 6MWT in subacute post-stroke patients. In this context, it seems that regular PA has beneficial physical and psychological effects for subacute post-stroke patients. In addition, the accelerometer incentive method allowed us to observe that our PA program was effective insofar as it encouraged patients to improve their PA status/intensity.

In view of the effect of our program and with the aim of improving the management of post-stroke patients in the subacute phase, PA education workshops should be implemented in hospital management.

## Figures and Tables

**Figure 1 ijerph-20-05908-f001:**
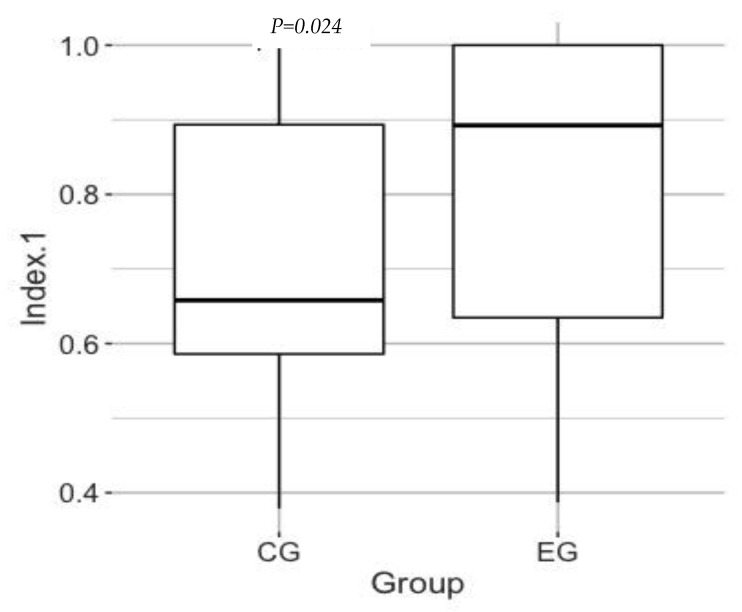
Comparison of the EQ-5D-5L at T1 between the 2 groups.

**Table 1 ijerph-20-05908-t001:** Baseline (T0) subject characteristics (Median, IQR).

Characteristics		*n* = 83
Mean age (years) ± SD		62.2 ± 13.6
Gender (*n* = ; M/F)		56/27
Mean time post-stroke (days) ± SD		77.9 ± 45.1
BMI (kg.m^−2^)		26.1 (5.4)
Nature of the stroke (%)	Ischemic	62 (75)
	Hemorrhagic	21 (25)
Location of the stroke (%)	Cerebellar	6 (7)
	Parietal	1 (1)
	Middle cerebral	54 (65)
	Anterior brain	4 (5)
	Posterior cerebral	3 (4)
	Brain stem	14 (17)
	Ventricle	1 (1)
Side reached (%)	Right	38 (46)
	Left	45 (54)
Background (*n* = /83)	Smoker	12
	HBP	40
	Diabetes	7
	Depression	4
	Heart disease	15
Mean Blood pressure (mm/Hg) ± SD	Systolic	137.6 ± 13.5
	Diastolic	80.1 ± 7.5

M: male, F: female; BMI: body mass index; SD: standard deviation, *n*: number of patients, %: percentage, HBP: High blood pressure.

**Table 2 ijerph-20-05908-t002:** Functional and physical parameters of the population at T0.

Parameters	Median	Lower Quartile	Superior Quartile	Rank (Min/Max)
Barthel Index (/100)	100	95	100	55/100
mFAC (/8)	6	6	8	2/8
MDI (/100)	94	77	100	32/100
6MWT (m)	358	270	485	30/658
EQ-5D-5L index	0.760	0.575	0.875	338/1000
Number Step	4081	1453	6153	33/16,084
TEE (Kcal)	1680	1482	1953	850/2415
TAEE (Kcal)	444	263	653	2.8/1049.3

SD: standard deviation; mFAC: Modified functional ambulation categories; MDI: Motricity Demeurisse Index; 6MWT: 6-min walk test; TEE: total energy expenditure; TAEE: Total active energy expenditure; Min: minimum; Max: maximum.

**Table 3 ijerph-20-05908-t003:** Distribution of EQ-5D-5L dimension responses at baseline (T0) and after the program (T1).

	Total Group (*n* = 83)		Experimental Group (*n* = 42)		Control Group (*n* = 41)	
Dimensions	T0	T1	T0	T1	T0	T1
**Mobility; *n* (%)**	***p* = 0.01**		***p* = 0.23**		***p* = 0.60**	
No problems	50 (60.2)	54 (65.1)	27 (64.3)	32 (76.2)	23 (56.1)	22 (53.7)
Slight problems	33 (39.8)	28 (33.7)	15 (35.7)	10 (23.8)	18(43.9)	18 (43.9)
Moderate problems	0 (0)	1 (1.2)	0 (0)	0 (0)	0 (0)	1 (2.4)
Severe problems	0 (0)	0 (0)	0 (0)	0 (0)	0 (0)	0 (0)
Unable to walk about	0 (0)	0 (0)	0 (0)	0 (0)	0 (0)	0 (0)
**Self-care; *n* (%)**	**NS**		***p* = 0.14**		***p* = 0.71**	
No problems	61 (73.4)	68 (81.9)	31 (73.8)	35 (83.3)	30 (73.2)	353 (80.5)
Slight problems	14 (16.9)	13 (15.7)	5 (11.9)	6 (14.3)	10 (24.4)	7 (17.1)
Moderate problems	8 (9.6)	2 (2.4)	6 (14.3)	1 (2.4)	1 (2.4)	1 (2.4)
Severe problems	0 (0)	0 (0)	0 (0)	0 (0)	0 (0)	0 (0)
Unable to wash or dress	0 (0)	0 (0)	0 (0)	0 (0)	0 (0)	0 (0)
**Usual activities; *n* (%)**	***p* = 0.03**		***p* = 0.10**		***p* = 0.97**	
No problems	33 (39.8)	46 (55.4)	19 (45.2)	26 (61.9)	19 (46.3)	20 (48.8)
Slight problems	39 (46.9)	32 (38.6)	20 (47.6)	26 (38.1)	17 (41.5)	16 (39.0)
Moderate problems	11 (13.3)	5 (6)	3 (7.2)	0 (0)	5 (12.2)	5 (12.2)
Severe problems	0 (0)	0 (0)	0 (0)	0 (0)	0 (0)	0 (0)
Unable to do usual activities	0 (0)	0 (0)	0 (0)	0 (0)	0 (0)	0 (0)
**Pain/discomfort; *n* (%)**	***p* = 0.01**		***p* = 0.42**		***p* = 0.68**	
No pain/discomfort	31 (37.3)	45 (54.2)	20 (47.6)	26 (61.9)	19 (46.3)	18 (46.3)
Slight pain/discomfort	42 (50.6)	32 (38.6)	19 (45.2)	14 (33.3)	20 (48.8)	18 (43.9)
Moderate pain/discomfort	10 (12.1)	6 (7.2)	3 (7.2)	2 (4.8)	2 (4.9)	4 (9.8)
Severe pain/discomfort	0 (0)	0 (0)	0 (0)	0 (0)	0 (0)	0 (0)
Extreme pain/discomfort	0 (0)	0 (0)	0 (0)	0 (0)	0 (0)	0 (0)
**Anxiety/depression; *n* (%)**	***p* = 0.01**		***p* = 0.21**		***p* = 0.51**	
Not anxious/depressed	26 (31.3)	46 (55.4)	17 (40.5)	25 (59.5)	16 (39.0)	20 (48.8)
Slightly anxious/depressed	49 (59.1)	35 (42.2)	24 (57.1)	17 (40.5)	22 (53.7)	20 (48.8)
Moderately anxious/depressed	8 (9.6)	2 (2.4)	1 (2.4)	0 (0)	3 (7.3)	1 (2.4)
Severely anxious/depressed	0 (0)	0 (0)	0 (0)	0 (0)	0 (0)	0 (0)
Extremely anxious/depressed	0 (0)	0 (0)	0 (0)	0 (0)	0 (0)	0 (0)

NS: not significant.

**Table 4 ijerph-20-05908-t004:** Evolution of the distance covered by the 6MWT and the functional parameter.

	EG; *n* = 42		CG; *n* = 41		*p*		
	T0	T1	T0	T1	G	T	G × T
6MWT	361.9 ± 148.4	430.8 ± 145.1	379.9 ± 147.8	391.6 ± 152.5	0.4	**0.01**	**0.01**
BI	94.8 ± 9.6	97.2 ± 6.5	94.7 ± 8.5	96.1 ± 9.9	0.7	0.2	0.5
MDI	90.4 ± 11.9	93.6 ± 10.1	85.0 ± 17.7	89.6 ± 15.2	0.05	0.1	0.7
mFAC	6.5 ± 1.6	7.2 ± 0.8	6.6 ± 1.2	6.5 ± 1.5	0.25	0.17	**0.03**

Mean ± SD; mFAC: Functional Ambulation Categories; EG Experimental group; CG: Control group; IB: Barthel Index; MDI: Motor Demeurisse Index; T: Time; G: Group; T0: pre-program; T1: 6-month measurement.

## Data Availability

The data presented in this study is available on request from the corresponding author. The data are not accessible to the public as they are health data. They are stored on a secure server at the University Hospital of Limoges.

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
