# Peer review of "Effect of Individualized Coaching at Home on Quality of Life in Subacute Stroke Patients"

_ijerph, 2023, doi:10.3390/ijerph20105908_

Round 1

Reviewer 1 Report

Thank you for the opportunity to review this article. The aim of the authors was to evaluate the effect of a home-based (HB) physical activity (PA) incentive program in quality of life (QoL) of patients in the subacute phase of stroke recovery.

The article is clearly written, despite some typos. The authors have clearly described the intervention methods in an easily understandable way, and the purposes are also easily deducible. However, I believe that the presentation of data at baseline, and its processing can be improved, to improve the quality of the product:

-      In Table 1, I encourage the authors to more clearly subdivide Nature and Location of the stroke. As they are presented, they are difficult to distinguish. Also, it would be helpful to have more information about the patients. The authors have included the value of BMI, but it would also be interesting to know the presence of comorbidities, habits concerning smoking, etc.

-     In Table 2, data collected at baseline are presented without distinction according to the patients' condition. Since it is noticeable that there are important differences between subjects (e.g., patients walking 30 or 658 meters in the 6MWT show a very different condition from each other), I encourage the authors to make a subdivision of the subjects according to their characteristics (age, clinical history, functional capacity?) in order to obtain more homogeneous categories.

-   Consequently, I believe that the results in Table 3 and Table 4 could be presented differently, highlighting how the intervention model proposed by the authors has different effects, depending on patient characteristics.

In conclusion, I believe that the article has the potential to deserve publication, but that more precision is needed in the description of the sample, and in the analysis and interpretation of the data collected.

Author Response

Thank you for the opportunity to review this article. The aim of the authors was to evaluate the effect of a home-based (HB) physical activity (PA) incentive program in quality of life (QoL) of patients in the subacute phase of stroke recovery.

The article is clearly written, despite some typos. The authors have clearly described the intervention methods in an easily understandable way, and the purposes are also easily deducible. However, I believe that the presentation of data at baseline, and its processing can be improved, to improve the quality of the product:

-      In Table 1, I encourage the authors to more clearly subdivide Nature and Location of the stroke. As they are presented, they are difficult to distinguish. Also, it would be helpful to have more information about the patients. The authors have included the value of BMI, but it would also be interesting to know the presence of comorbidities, habits concerning smoking, etc.

Thank you for this first comment. We have made a number of changes to the table to make it more readable, including the nature and location of the stroke. We have added the main antecedents that we observed in our participants.

-     In Table 2, data collected at baseline are presented without distinction according to the patients' condition. Since it is noticeable that there are important differences between subjects (e.g., patients walking 30 or 658 meters in the 6MWT show a very different condition from each other), I encourage the authors to make a subdivision of the subjects according to their characteristics (age, clinical history, functional capacity?) in order to obtain more homogeneous categories.

Thank you for your comment. This is a point that we had discussed with the other authors. However, we did not want to establish a category at this stage. Indeed, in the rest of the manuscript there were  two groups, experimental and control. The main objective of this work is to evaluate the effect of the programme on the patients' quality of life. If we subdivide the table, which is only intended to present the baseline participants, we are afraid of losing the readers, especially when it comes to distinguishing between the two groups. We prefer to leave it that way. Furthermore, we do not think that this addition adds any value to our article.

 -   Consequently, I believe that the results in Table 3 and Table 4 could be presented differently, highlighting how the intervention model proposed by the authors has different effects, depending on patient characteristics.

In conclusion, I believe that the article has the potential to deserve publication, but that more precision is needed in the description of the sample, and in the analysis and interpretation of the data collected.

Thank you for the further comments on our two tables. I would like to clarify again that these are the values for our entire baseline sample. We have had an exchange with the biostatistician of our study. According to him it is not recommended to subdivide our sample. Our study was a randomised, controlled, single-blind study. The separation of the two groups was done according to the inclusion and exclusion criteria. We did not perform a comparison between the two baseline groups as this is equivalent to testing chance.  That makes no statistical sense. Subdividing the two groups according to the characteristics of the individuals, which as you have seen are very heterogeneous, would cause us to lose a lot of statistical power.  Furthermore, by subdividing groups of 42 participants, our sample would be too small to draw  any conclusions.

The major conclusion of this work clearly shows that an improvement in Quality of Life is observed after the programme in the experimental group, which has the same characteristics as the control group.

If we have  not understood  your question, please do not hesitate to contact us.

Reviewer 2 Report

-The most recent bibliographic citation dates from 2018. There are many studies after that date and related to the topic addressed by the article.

- It would be interesting to know the physical activity program that was designed for each participant, at least generically, to describe what it was based on.

Author Response

-The most recent bibliographic citation dates from 2018. There are many studies after that date and related to the topic addressed by the article.

Thank you for this comment and indeed we have changed several references that were a bit old

- It would be interesting to know the physical activity program that was designed for each participant, at least generically, to describe what it was based on.

Thank you very much for your comment. We have provided some additional information as it seems that we were not very clear. As part of  overall care, all patients in the department receive advice on physical activity, nutrition, etc. They are directed toward suitable sports activities. When they leave the hospital all patients benefit from the usual care, with a follow-up in physiotherapy. The incentive programme is additional for patients in the experimental group. As you will have understood, it is not a standardised programme with intensities to be respected, a frequency of exercise... It is a question of accompanying them in an active lifestyle respecting the recommendations for activity

We have made changes:

Following inclusion, a physical activity (PA) advice program was designed and implemented by a PA therapist for all eligible participants . In the French Society of Physical Medicine and Rehabilitation and the French Neurovascular Society’s PA recommendations, the benefits of an active lifestyle in preventing stroke recurrence are emphasized. The presentation included an educational diagnosis, selection of PA, and definition of goals. Usual care was provided for all eligible participants for 12 months after hospital discharge, including outpatient therapy and medical appointments at 1, 6, and 12 months. The EG underwent an incentive coaching program, which involved monitoring, weekly phone calls, and a home visit every 3 weeks for 6 months. An activity tracker (AT) (SenseWear Armband, BodyMedia, Pittsburgh, PA, USA) was used to monitor PA at home. The participants were asked to wear the AT when they woke up and to remove it before going to bed. This sensor was selected due to its frequent utilization in activity analysis and its unique feature of activation upon contact with the wearer's skin. Therefore, the data obtained correspond to AT wearing time. The EG participants were routinely monitored through completion of a daily chart to measure subjective perception of PA, weekly phone calls to encourage regular PA and to inquire about the PA measuring device, and home visits every three weeks to receive participant feedback on PA level (step counts [SC]) and set objectives for the next visit. The aim of this programme was to encourage participants to meet the PA recommendations, without offering them specific sessions but rather encouraging them to maintain or have an active lifestyle.

Reviewer 3 Report

I have read the paper and make the following cmments:

  • Words like ‘however’ or ‘moreover’ never truly add anything and are superfluous and should be removed as per, “ However, the benefits of physical activity”, taken from the abstract or “ However, the benefits of PA on QOL” taken form the introduction and  “ Moreover, fatigue and Quality of Life” taken from the introduction
  • Acronyms like ‘PA’ should be s[pelt out the first time they are iused, even in the abstract, as per, “home-based PA incentive program”.
  • ~80 days post stroke is not long enough to be sufficiently critical and such short follow up needs explaining.
  • It may simply be that the experimental group had greater contact with the team than did the control group and hence it may have nothing to do with the actual home course.
  • The syntax in the following sentence is wrong and needs correcting, Follows on the one hand, the authors who studied the durability of these benefits 43 showed that they were not maintained… “
  • “medicalized” is NOT a word and should be changed as per, “carried out in medicalized structures “, taken from the introduction.
  • In the following quotation, ‘interesting’ means absolutely nothing and should be changed, “… becoming an interesting strategy for dealing…”, taken from the introduction.
  • When the author(s) state ‘non-inclusion’, as per, “The non-inclusion criteria were physical disorders” do they actually mean ‘exclusion criteria”?
  • To say that the control group ‘may have’ received implies that no all did and that is important as it may not be the programme of PA that was therapeutic but merely the attention received by the patients, as per “usual care for 12 months, which 99 may have included outpatient therapy, medical appointments at 1, 6 and 12 months. The 100 EG participants underwent the individualized coaching program”, taken from the methods.
  • In the following quotation, what does aqs mean? “This sensor aqs choose because it is regularly used”. Also the syntax and grammar are wrong.
  • Words like ‘therefore’ are similar to ‘however’ and ‘moreover’ and can be omitted, as per, “Therefore, the data obtained correspond to AT”, taken form the methods.
  • I will refrain from further analysing the English, syntax and grammar but argue that the paper needs revisiting to correct these minor issues which do detract from the presentation.
  • The following sentence, taken from the discussion, makes little sense in its current form, “Unlike many studies proposing PA programs for post-stroke patients, which are generally standardized, our program was essentially based on incentive 240 them to increase their level of daily activity.”
  • “A recruitment bias was detected during inclusion” is very important as low educational groups declined entry into the trial which again suggests that it might have been the bincreased attention rather than the PA which was instrumental in the improvement provided.
  • The authors make a similar statement in their identification of the second limitation to their investigation in that they did not want the second , namely control group to be awar that they were bewing studied. This reinforces the argument that is may NOT have been the PA that was effective but the simple act of inclusion into a trial with closer monitoring. The authors have not commented on this at all and it may well be a fundamental flaw in the whole project.
  • In the light of the above comments the authors are encouraged to revisit their paper and correct for the langaugae problems as well as reconsidering their findings and what they mean.

Author Response

Dear Reviewer
I wanted to thank you for the comments you had on our article. I hope that we have been able to respond to them and that the article will be clearer as a result. I would also like to point out that we have had our article proofread by two English-speaking reviewers. I hope that the English will be of better quality. It is always a bit tricky, especially as the two other reviewers did not point out any major errors in our manuscript.

I remain at your disposal if needed

·       Words like ‘however’ or ‘moreover’ never truly add anything and are superfluous and should be removed as per, “ However, the benefits of physical activity”, taken from the abstract or “ However, the benefits of PA on QOL” taken form the introduction and  “ Moreover, fatigue and Quality of Life” taken from the introduction

Thank you for this comment. Indeed, these terms are very often superfluous. We have removed them

·       Acronyms like ‘PA’ should be s[pelt out the first time they are iused, even in the abstract, as per, “home-based PA incentive program”.

We have made the necessary changes starting in the summary

·       ~80 days post stroke is not long enough to be sufficiently critical and such short follow up needs explaining.

We have changed the sentence to make it clearer

·       It may simply be that the experimental group had greater contact with the team than did the control group and hence it may have nothing to do with the actual home course.

Thank you very much for your comment. We have provided some additional information as it seems that we were not very clear. As part of overall care, all patients in the department benefit from advice on physical activity, nutrition, etc. They are directed towards suitable sports activities. When they leave the hospital alle patients benefit from t usual care, with a follow-up in physiotherapy. The incentive programme is additional for patients in the experimental group. As you will have understood, it is not a standardised programme with intensities to be respected, a frequency of exercise... It is a question of accompanying them in an active lifestyle respecting the activity recommendations.

So you're quite right, it's interesting to see that if we accompany the patients we finally observe a better quality of life.

·       The syntax in the following sentence is wrong and needs correcting, Follows on the one hand, the authors who studied the durability of these benefits 43 showed that they were not maintained… “

Thank you for this comment. Indeed the sentence is very long and difficult to understand, so we have modified it

·       “medicalized” is NOT a word and should be changed as per, “carried out in medicalized structures “, taken from the introduction.

We have changed the sentence to make it clearer

·       In the following quotation, ‘interesting’ means absolutely nothing and should be changed, “… becoming an interesting strategy for dealing…”, taken from the introduction.

We have changed the word :

The literature shows that the establishment of support teams, which intervene at home after hospital discharge, is becoming a beneficial strategy for dealing with these types of difficulties

·       When the author(s) state ‘non-inclusion’, as per, “The non-inclusion criteria were physical disorders” do they actually mean ‘exclusion criteria”?

We have changed the sentence to make it clearer

Ethical approval was obtained from institutional committees (RCB: 2012-A01456-37) and potential participants with a first stroke who met the inclusion criteria (age ≥ 18 years; first ischemic or hemorrhagic stroke within <6 months; Modified Functional Ambulatory Categories (FAC) ≥2; registered with the French social security system) were screened. Exclusion criteria included pre- or post-stroke physical disorders limiting gait skills, cognitive disorders impeding comprehension of PA education

·       To say that the control group ‘may have’ received implies that no all did and that is important as it may not be the programme of PA that was therapeutic but merely the attention received by the patients, as per “usual care for 12 months, which may have included outpatient therapy, medical appointments at 1, 6 and 12 months. The EG participants underwent the individualized coaching program”, taken from the methods.

Thank you for your comment. You were not the only reviewer to alert us to this explanation. We have made several changes to address this point.

·       In the following quotation, what does aqs mean? “This sensor aqs choose because it is regularly used”. Also the syntax and grammar are wrong.

You are right. We have changed the sentence.

The EG underwent an incentive coaching program, which involved monitoring, weekly phone calls, and a home visit every 3 weeks for 6 months.

·       Words like ‘therefore’ are similar to ‘however’ and ‘moreover’ and can be omitted, as per, “Therefore, the data obtained correspond to AT”, taken form the methods.

We have made the change:

The data obtained correspond to AT wearing time.

·       I will refrain from further analysing the English, syntax and grammar but argue that the paper needs revisiting to correct these minor issues which do detract from the presentation.

Thank you for your feedback. Our document has already been proofread by an English speaker. However, following your comments we had  the corrections proofread again.

·       The following sentence, taken from the discussion, makes little sense in its current form, “Unlike many studies proposing PA programs for post-stroke patients, which are generally standardized, our program was essentially based on incentive them to increase their level of daily activity.”

You are right. We have changed the sentence.

·       “A recruitment bias was detected during inclusion” is very important as low educational groups declined entry into the trial which again suggests that it might have been the increased attention rather than the PA which was instrumental in the improvement provided.

Thank you for your comment. However, I think there is some confusion. The level of education was not assessed in our study. The bias we have is mainly related to the fact that patients have a high level of Autonomy, which we determined by the barthel Index.

Let me put the sentence in question again:

A recruitment bias was detected during inclusion. The participants were mainly patients with a high level of autonomy (BI of > 90 points) and motricity (mFAC > 6), and patients with very severe stroke do not wish to be included in this study.

·       The authors make a similar statement in their identification of the second limitation to their investigation in that they did not want the second , namely control group to be awar that they were bewing studied. This reinforces the argument that is may NOT have been the PA that was effective but the simple act of inclusion into a trial with closer monitoring. The authors have not commented on this at all and it may well be a fundamental flaw in the whole project.

You are absolutely right, and I consider that this limit has no reason to exist. I have therefore removed it. I have added a limit in connection with your various comments

Unlike the main paper of this work that we published (Mandigout et al, 2021) on the effects of the programme on the 6min walk perimeter, in this work it seems difficult to attribute the improvement in quality of life solely to the fact of physical activity. For this work we have to consider the whole programme, which also allowed the creation of a social link between the patient and the professionals who visited their homes every 3 weeks.

However, your comment made us realise that we had forgotten an important inclusion criterion. All patients included in the study (EG and CG) were followed by the Hemipass mobile team, a team that accompanies patients at home following stroke. Our programme was supplementary  to this service. You are right, the patients in the experimental group had more regular follow-up. However, the patients in the control group also received visits according to their needs and the medical follow-up requested by the doctor.

·       In the light of the above comments the authors are encouraged to revisit their paper and correct for the langaugae problems as well as reconsidering their findings and what they mean.

Round 2

Reviewer 1 Report

I thank the authors for the changes they made to the text, and for answering my doubts. I understand and agree with the choice not to focus on differences in participants' performance. However, I believe that the fact that the proposed program has positive effects on such a heterogeneous sample, in terms of performance, is a point of strength that would deserve to be emphasized by the authors.

There is a great need in the field of rehabilitation for "universal" intervention models that can be adaptable to patients with different levels of autonomy. For this reason, I invite the authors to include this consideration.

I would add two minor comments:

-        Line 94: I invite the authors to complete the sentence

-        Line 166: I encourage the authors to remove "HBP: High blood pressure," and relocate it into the legend, below Table.1

Author Response

Dear Professor

I wanted to thank you for these additional remarks. 
I have made the corrections in red in the text concerning the two minor requests:

-        Line 94: I invite the authors to complete the sentence Following inclusion, a PA advice program was designed and implemented by a PA therapist for all eligible participants according to the recommendations of the French Society of Physical Medicine and Rehabilitation and the French Neurovascular. The presentation included an educational diagnosis (benefits of an active lifestyle in preventing stroke recurrence), selection of PA, and definition of goals [15].   -        Line 166: I encourage the authors to remove "HBP: High blood pressure," and relocate it into the legend, below Table.1

M: male, F: female; BMI: body mass index; SD: standard deviation, N: number of patients, %: percentage, HBP: High blood pressure.

Concerning your first comment, I have taken the liberty of adding a small paragraph in order to better justify what we said. Indeed, in the main study we show no difference at T1 between the two groups on TM6. It is important to make this clear. Nevertheless, you are right, the result is very promising, especially as the programme is not really complicated to implement.
I think that this paragraph added to the conclusion should be enough to justify the interest of encouraging the practice of PA

The PA program increased the walking distance covered at the 6MWT by 11.7m in the CG and by 68.9m in the EG. The evolution of the performance at the 6MWT between T0 and T1, evaluated from an analysis of variance, differed between our two groups. Post hoc tests showed a significant 6MWT increase for the EG (18%, p<0.001) after the program. These results should be taken with great caution. Indeed, in our main study[21], we did not show a difference at T1 between the two groups when the statistical analysis was done on intention-to-treat. However, the primary endpoint was calculated based on a 30% increase in 6MWT. This result could not be achieved due to the low level of impairment of our programme participants (Barthel index > 94). Our results are in agreement with the study by Duncan et al[22]. This study proposed a home-based PA program, three times a week, with sessions of approximately 1.5 hours, for 12 weeks. At the end of the intervention, the PA program increased the walking distance at the 6MWT by 59 meters for the patients who followed the home program and by 35 meters for the control patients. We show in this secondary analysis that, despite a high level of heterogeneity and a low level of impairment, our incentive programme, consisting of individualized follow-up, significantly improved the quality of life and walking capacity of our participants.

I remain at your disposal for any further changes.

Yours sincerely

Reviewer 2 Report

Thank you very much for the aclarations. Congratulations for the work done.

Author Response

Many thanks